# NeuroRVQ: Multi-Scale EEG Tokenization for Generative Large Brainwave Models

## Abstract

Electroencephalography (EEG) captures neural activity across multiple temporal and spectral scales, yielding signals that are rich but complex for representation learning. Recently, EEG foundation models trained to predict masked signal-tokens have shown promise for learning generalizable representations. However, their performance is hindered by their signal tokenization modules. Existing neural tokenizers fail to preserve high-frequency dynamics, limiting their ability to reconstruct EEG signals with high fidelity. We introduce NeuroRVQ, a scalable Large Brainwave Model (LBM) centered on a codebook-based tokenizer. Our tokenizer integrates: (*i*) multi-scale feature extraction modules that capture the full frequency neural spectrum; (*ii*) hierarchical residual vector quantization (RVQ) codebooks for high-resolution encoding; and, (*iii*) an EEG signal phase- and amplitude-aware loss function for efficient training. This design enables efficient EEG compression while supporting accurate reconstruction across all frequency bands, leading to robust generative masked modeling. Our empirical results demonstrate that NeuroRVQ achieves lower reconstruction error and outperforms existing LBMs on a variety of downstream tasks. More broadly, the NeuroRVQ tokenizer establishes a strong prior for codebook-based general-purpose brainwave models, enabling advances in neural decoding, generative modeling and multimodal biosignal integration.

## 1 Introduction

Brain-Computer Interface (BCI) systems enable direct communication between the brain and the external world by analyzing brainwaves recorded by Electroencephalography (EEG) devices. EEG signals can represent the full spectrum of human experience, from sleep and emotions to movement. Decoding EEG signals is a fundamental component of BCIs that find application in various areas such as emotion recognition (Torres et al., 2020; Xu et al., 2018), epileptic seizure detection (Alkawadri, 2019; Djoufack Nkengfack et al., 2021) and robotic control (Irimia et al., 2012).

Brain activity unfolds across multiple frequency scales. Brainwave signals can be categorized into distinct frequency bands—namely, delta (0.5–4 Hz), theta (4–8 Hz), alpha (8–13 Hz), beta (13–30 Hz) and gamma (>30 Hz)—each linked to specific cognitive and physiological states. For example, delta waves dominate during deep sleep, while alpha waves emerge during relaxing states (especially with closed eyes) but tend to decrease in amplitude during movement. Together, these bands provide a dynamic window into the brain's activity and internal states. *EEG analysis must capture the multiscale structure of brainwaves, both slow and fast dynamics, to fully represent neural information.*

A range of techniques, from advanced signal processing to machine learning, have been employed for brainwave decoding. More recently, Large Brainwave Models (LBMs) offer strong generalization capabilities without relying on BCI task-specific data collection and model training. These models are typically trained using self-supervised masked learning: EEG signals are discretized into *tokens*; and, during training, tokens are intentionally masked so that models learn to reconstruct them. However, reconstructing raw EEG signals remains a significant challenge due to their inherently noisy and complex nature. Tokenizers must effectively capture spatio-temporal patterns (e.g., location of recording electrodes, time and frequency) to convert them into meaningful token representations. *Current tokenizers struggle to preserve the complete structural information (especially high frequency components) necessary for robust generative masked modeling.* The key to unlocking foundation-scale masked modeling for EEG lies in the tokenizer:

> A well-designed tokenizer should not only compress continuous neural signals into discrete tokens but also enable faithful reconstruction of the original waveform across all important frequency scales.

In this work, we introduce NEURORVQ, an efficient and scalable LBM architecture. The main contribution of this work is its state-of-the-art tokenizer that:

(1) captures multi-scale frequency features by applying temporal convolutions with varying kernel sizes;

(2) encodes hierarchical representations in Residual Vector Quantization (RVQ) codebooks, one per frequency scale, enabling us to capture the complex patterns necessary for high-fidelity signal reconstruction in 32 codebooks ($2^{32}$ parameters);

(3) employs a training procedure grounded in strong signal processing principles that captures both amplitude and wrapped phase information of the EEG signal via sine and cosine representations.

We evaluate the efficacy of our tokenizer by comparing NEURORVQ with state-of-the-art LBMs. NEURORVQ achieves up to 15% higher accuracy on four BCI classification tasks, demonstrating the effectiveness of codebook-based modeling when codebooks faithfully reconstruct brain signals.

## 2 BACKGROUND & RELATED WORK

The roots of EEG analysis can be traced back to classical neuroscience methods, when handcrafted features such as Power Spectral Density (PSD) bands and Independent Component Analysis (ICA) were regarded as standard (Bashashati et al., 2007; Handy, 2009; Rao, 2013; Nam et al., 2018; McFarland et al., 2006). While these methods provided valuable insights into neural dynamics, they often failed to generalize due to high inter-subject variability, the noisy nature of EEG signal and their limited capacity to capture the full complexity of brain activity.

The deep learning era marked a shift towards data-driven feature extraction: brainwave decoders such as EEGNet (Lawhern et al., 2018), EEGInception (Santamaría-Vázquez et al., 2020), EEGConformer (Song et al., 2023), and Brainware-Scattering Net (Barmpas et al., 2023) reduced reliance on handcrafted features by automatically learning representations of brain activity. To address domain-specific challenges such as subject variability, researchers incorporated domain-specific knowledge into these architectures, such as subject-alignment to adjust feature representations (Bakas et al., 2022; Wei et al., 2022; Barmpas et al., 2024b). Despite these advances, most deep learning models remain limited: they require carefully labeled data and extensive task-specific training, hindering their ability to scale and generalize across diverse BCI tasks.

More recently, the success of large foundation models in natural language processing (Brown et al., 2020; Touvron et al., 2023) and computer vision (Mizrahi et al., 2023) has motivated their adoption in EEG decoding. LaBraM (Jiang et al., 2024), NeuroGPT (Cui et al., 2024), CBraMod (Wang et al., 2025), EEGPT (Wang et al., 2024) and BIOT (Yang et al., 2023) are examples of foundation models for brain signals, pretrained on massive unlabeled datasets using self-supervised signal reconstruction objectives. These models represent a promising direction for capturing complex signal patterns beyond task-specific methods. However, they face a fundamental bottleneck—signal tokenization.

Tokenization is critical for robust generative masked modeling, yet current tokenizers struggle to capture the full structure of brain signals. NeuroGPT relies on a deep backbone-based tokenizer while other models like BIOT and EEGPT on transformer-based tokenizers, which often learn entangled and redundant representations not optimized for reconstruction. As a result, the generated tokens lack the fidelity required for masked prediction. In contrast, LaBraM adopts a codebook-based approach to learn reconstructable neural tokens. However, while discrete codebook-based tokenizers such as VQ-VAE (Esser et al., 2020) work well in computer vision, directly adopting them for brain signals have not yielded faithful signal reconstruction—until NEURORVQ.

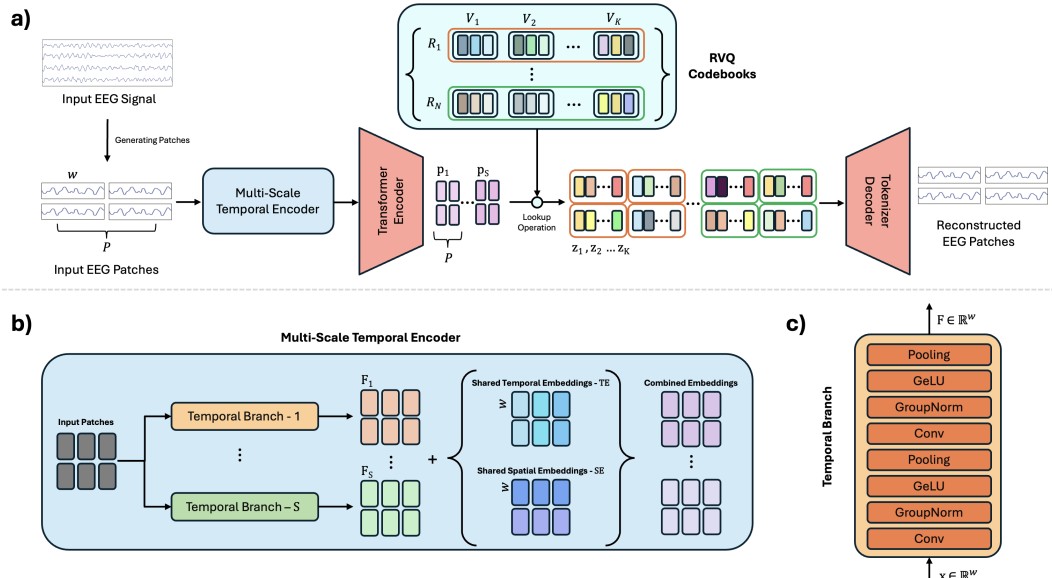

Figure 1: **a)** The NEURORVQ tokenizer model architecture; **b)** the multi-scale temporal encoder architecture; and **c)** the temporal branch architecture. The multi-scale temporal encoder extracts multi-scale temporal features that are passed to the transformer encoder. RVQ codebooks per temporal branch discretize the EEG patches into neural tokens. The tokenizer decoder reconstructs the EEG signal by using the Fourier spectrum.

## 3 NEURORVQ TOKENIZER

### 3.1 MODEL ARCHITECTURE

NEURORVQ tokenizer is a specialized network designed to convert raw EEG signals into a sequence of compact and informative neural tokens. This transformation reduces the inherently high-dimensional and noisy nature of EEG into a structured lower-dimensional representation that preserves essential temporal–spectral patterns. In doing so, the tokenizer provides a kind of "neural grammar" for brain activity.

Figure 1 illustrates the tokenizer architecture and processing flow. The input multi-variate timeseries is first segmented into EEG patches. These patches are encoded by the multi-scale temporal encoder, that captures features in multiple resolutions and are then combined via the transfromer encoder. For each scale, RVQ codebooks discretize the embeddings into a sequence of neural tokens. Finally, these tokens are combined and passed through the tokenizer decoder to reconstruct the input EEG patches using the Fourier spectrum.

**Generating Patches:** Let $X \in \mathbb{R}^{C \times T}$ denote the input EEG signal, where $T$ is the number of time points and $C$ is the number of electrodes. The signal is first segmented into temporal patches. To ensure that the tokenizer can accommodate EEG signals with arbitrary numbers of channels and variable durations, we adopt the following approach: during model pre-training, each input multi-variate EEG signal is represented by $P$ patches of length $w$ (corresponding to a 1-second time window). This results in a segmented input sample $\mathbf{x} \in \mathbb{R}^{P \times w}$ which provides flexibility to handle heterogeneous EEG recordings and facilitates robust pre-training across diverse datasets.

**Multi-Scale Temporal Encoder:** The first step is to extract multi-scale temporal features from each input patch using an inception-style module with $S$ distinct temporal scales. To achieve this, we apply 1-D temporal convolutions with varying kernel sizes $K_{\text{temporal}_1}, K_{\text{temporal}_2}, \ldots, K_{\text{temporal}_S}$. Each temporal branch consists of the following sequence of operations: a 1-D convolution, followed by group normalization, GELU activation and pooling (this sequence of operations is repeated twice). This design enables the network to capture temporal features at multiple scales resulting in $S$ outputs $F_1, F_2, \ldots, F_S$ where $F_i \in \mathbb{R}^w$ where $w$ is the patch time window length.

**Transformer Encoder:** Since the tokenization of EEG signal takes place in a per-patch level, the vital temporal and spatial information is incorporated through the use of trainable temporal $TE$ and spatial $SE$ embeddings, each with embedding dimension $w$. $SE = \{se_1, ..., se_{|\mathcal{C}|}\}$ is indexed by the positions of the patches' electrodes within a list of all electrodes in the entire pre-training database $\mathcal{C}$ and $TE = \{te_1, ..., te_P\}$ since each input consists of $P$ patches of length $w$. These embeddings are added to the extracted multi-scale features $F_1, F_2, \ldots, F_S$ and are shared across all $S$ temporal branches. The combined embeddings of each temporal branch are then passed through a series of shared transformer layers (Vaswani et al., 2023) resulting in the multi-scale patch representations $\mathbf{p}_1, \mathbf{p}_2, \ldots, \mathbf{p}_S \in \mathbb{R}^D$. The shared transformer layers have incorporated the modification of Dehghani et al. (2023).

**RVQ Codebook:** To efficiently discretize the multi-scale patch representations $\mathbf{p}_1, \mathbf{p}_2, \ldots, \mathbf{p}_S$ while preserving fine-grained information of the EEG signal, we use $S$ Residual Vector Quantization (RVQ) codebooks. For each temporal branch, the RVQ codebook $\mathcal{R}$ is defined as:

$$\mathcal{R} = \{\mathcal{V}_i | i = 1, ..., N\} \tag{1}$$

where $\mathcal{V}_i$ is a single codebook of neural tokens defined as:

$$\mathcal{V}_i = \{\mathbf{v}_j | j = 1, ..., K\} \in \mathbb{R}^{K \times D} \tag{2}$$

where $K$ is the number of discrete neural tokens and $D$ is the dimensionality of each token. The discretization of the patch representations is performed iteratively. For each $\mathcal{R}$, at step 1, a quantizer maps each patch representation $\mathbf{p}^1$ to its corresponding neural token $\mathbf{v}^1 \in \mathcal{V}_1$. Specifically, for each patch, the quantizer selects the nearest neighbor $\mathbf{z_1}$ from the codebook $\mathcal{V}_1$:

$$\mathbf{z_1} = \arg \min_{\mathbf{v} \in \mathcal{V}_1} \left\| l_2(\mathbf{p}^1) - l_2(\mathbf{v}) \right\| \tag{3}$$

At each next step $i + 1$, the residual patch representation $\mathbf{p}^{i+1}$ for the next codebook $\mathcal{V}_{i+1}$ is updated by subtracting the selected token $\mathbf{z}_i$:

$$\mathbf{p}^{i+1} = \mathbf{p}^i - \mathbf{z}_i \tag{4}$$

This process is repeated for all $N$ codebooks, producing a sequence of tokens $\mathbf{z}_1, \mathbf{z}_2, \ldots, \mathbf{z}_N$ for each temporal branch separately. The final reconstructed patch representations $\hat{\mathbf{p}}$ is obtained by summing the contributions of all selected tokens $\mathbf{z}_1, \mathbf{z}_2, \ldots, \mathbf{z}_N$:

$$\hat{\mathbf{p}} = \sum_{i=1}^{N} \mathbf{z}_i \tag{5}$$

At the end of this module, we have $\hat{\mathbf{p}}_1, \hat{\mathbf{p}}_2, \ldots, \hat{\mathbf{p}}_S$ corresponding to the $S$ temporal branches.

**Tokenizer Decoder:** The purpose of the decoder module (a series shared transformer layers followed by prediction heads) is to reconstruct the original signal information based on the learned codebook tokens. Directly reconstructing raw EEG signals has been shown to yield unstable training and poor performance, due to the inherently noisy and high-dimensional nature of EEG. Following the strategy of LaBraM (Jiang et al., 2024), we instead leverage the Fourier spectrum (more specifically the phase $\phi$ and spectral amplitude $A$) as a more structured target for reconstruction.

### 3.2 TOKENIZER TRAINING

Reconstructing the phase $\phi$ directly with mean squared error (MSE) is suboptimal due to the periodic nature of the Fourier phase (see Appendix A). To address this, our architecture replaces raw phase reconstruction with a sine/cosine phase representation and we predict the $\sin \hat{\phi}$ and $\cos \hat{\phi}$, which are smooth, continuous and better suited as loss terms. For the amplitude component, rather than reconstructing raw spectral magnitudes $A$, we operate in the log-amplitude domain $\log(1 + A)$. This logarithmic scaling compresses the large dynamic range of EEG spectra and places relatively greater emphasis on reconstructing high-frequency components, a key weakness in existing tokenizers.

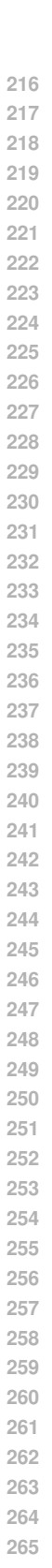

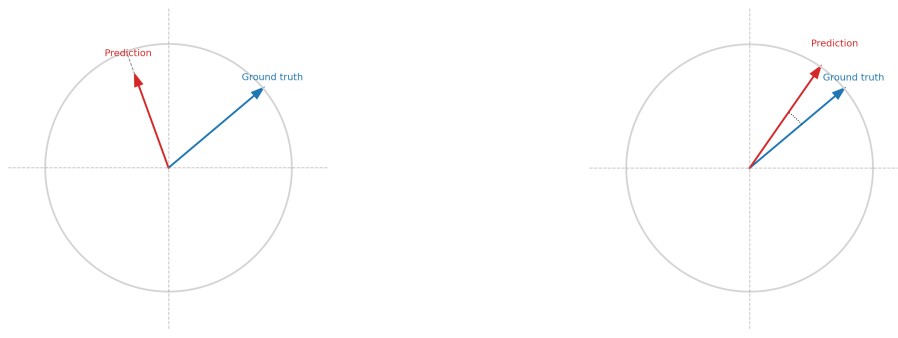

(a) Prediction not constrained to unit circle.  (b) Prediction constrained to $|v| = 1$.

Figure 2: Comparison of phase modeling approaches: (a) Independent MSE can produce invalid predictions (b) Unit-circle-aware loss enforces valid phase angles.

Therefore, the tokenizer decoder has three prediction heads responsible for reconstructing $\log(1 + \hat{A})$, $\sin \hat{\phi}$, and $\cos \hat{\phi}$, respectively.

Naively treating the sine and cosine components independently by applying mean squared error to each dimension separately during training fails to explicitly capture the circular nature of phase: minimizing the MSE of sine and cosine individually does not guarantee that the resulting vector lies on the unit circle and it may allow predictions that deviate from valid phase angles. To address these limitations, we introduce a unit-circle-aware phase loss $\mathcal{L}_{\text{unit-loss}}$ that directly optimizes the directional alignment of the predicted phase vector with the target phase while enforcing unit-norm constraints. Specifically, we represent each phase as a 2D vector $(\cos \phi, \sin \phi)$ and compute the cosine similarity between predicted and target vectors, combined with a small regularization term that penalizes deviations from unit length:

$$\mathcal{L}_{\text{unit-loss}} = 1 - \underbrace{\sum_i \frac{\cos \hat{\phi}_i \cos \phi_i + \sin \hat{\phi}_i \sin \phi_i}{\sqrt{\cos \hat{\phi}_i^2 + \sin \hat{\phi}_i^2} \sqrt{\cos \hat{\phi}_i^2 + \sin \hat{\phi}_i^2}}}_{\text{Cosine Similarity}} + \underbrace{\lambda_{circle} \cdot \sum_i \left( \cos \hat{\phi}_i^2 + \sin \hat{\phi}_i^2 - 1 \right)^2}_{\text{Unit-Circle Penalization Term}}$$

(6)

This formulation has several advantages: it naturally respects the circular topology of phase, avoids discontinuities near angular boundaries and ensures that all predicted vectors correspond to valid phase angles. The phases are essentially represented as unit vectors in 2D space, making the errors between them more geometrically meaningful.

Finally, although reconstructing raw EEG waveforms is unstable, we found that incorporating a temporal-domain reconstruction loss provides additional guidance. Specifically, we transform the predicted log-amplitude and phase back to the time domain (the predicted log-amplitudes are exponentiated and rescaled to recover the original amplitude spectrum before inverse Fourier transformation) and compute a temporal MSE term. The NEURORVQ tokenizer is thus trained end-to-end with the following objective:

$$\mathcal{L}_T = \sum_i \underbrace{\left\| \log(1 + \hat{A}_i) - \log(1 + A_i) \right\|_2^2}_{\text{Log-Amplitude Loss}} + \mathcal{L}_{\text{unit-loss}} + \underbrace{\left\| \hat{X}_i - X_i \right\|_2^2}_{\text{Temporal Loss}} + \mathcal{L}_Q,$$

(7)

where $\hat{A}$, $\sin \hat{\phi}$, and $\cos \hat{\phi}$ denote reconstructed Fourier components, $\hat{X}$ denotes reconstructed EEG in the time domain and $\mathcal{L}_Q$ is the quantization loss as defined in Esser et al. (2020).

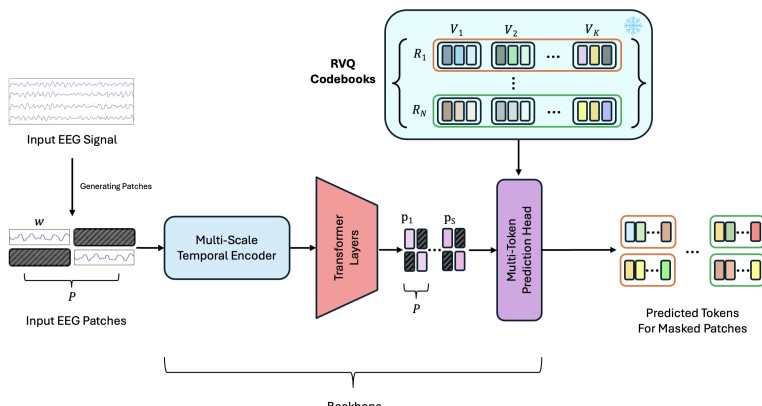

Figure 3: NEURORVQ Foundation Model. During pre-training, random EEG patches are masked, and the model is trained to reconstruct the missing tokens from the RVQ codebook using the surrounding visible patches as context.

## 4 NEURORVQ FOUNDATION MODEL

NEURORVQ is a scalable foundation model that operates on the tokenized representation. By working at the token level rather than raw signals, this model can better capture long-range dependencies, learn abstract neural dynamics and enable efficient pretraining across diverse EEG datasets. The model leverages the learned codebooks $\mathcal{R}$ from the tokenizer stage and is trained using a masked-token prediction strategy, where a subset of input patches is randomly masked. This objective encourages the network to infer missing tokens from their surrounding context.

Figure 3 shows the architecture of NEURORVQ foundation model. The architecture consists of a multi-scale temporal encoder (Section 3.1) that extracts hierarchical temporal features across multiple resolutions, followed by a series of transformer layers that model long-range complex dependencies. A series of token prediction heads reconstructs the masked tokens from the contextualized embeddings using the trained $\mathcal{R}$ codebooks, which are used only during the pre-training of the backbone model. In line with LaBraM, a symmetric masking strategy is employed, generating the spatially inverse of each original mask during training to enhance efficiency and data diversity. By combining hierarchical temporal encoding, transformer-based context modeling and symmetric masking, the foundation model effectively captures rich spatio-temporal patterns and produces high-fidelity token predictions. For downstream tasks, the pre-trained backbone is extended with task-specific classifications heads.

## 5 EXPERIMENTS

We evaluate NEURORVQ along two dimensions (*i*) the quality of its tokenizer, compared with the tokenizer of LaBraM, the current state-of-the-art codebook-based method for signal reconstruction; and (*ii*) the performance of the NEURORVQ foundation model, compared with other pre-trained models, fine-tuned for downstream classification tasks.

### 5.1 EXPERIMENTAL SETUP

**Hyperparameters / Training Process:** NEURORVQ was trained using the settings described in Appendix B. The tokenizer was trained for 100 epochs with $S = 4$ temporal branches and 4 $\mathcal{R}$ RVQ codebooks each consisting of 8 single codebooks $\mathcal{V}_i \in \mathbb{R}^{8192 \times 128}$ and $\lambda_{circle} = 0.4$. The foundation model was trained for 50 epochs in the same datasets. The experiments were run on NVIDIA DGX with 4 NVIDIA Tesla V100 GPUs.

**Datasets:** We have used 13 large-scale EEG datasets, consisting of 12 public and 1 self-collected ($\sim$235 hours of our own self-collected motor data) datasets (we refer the reader to Appendix D). All datasets were resampled to the same sampling frequency of 200Hz.

## 5.2 TOKENIZER EVALUATION

The first step involves comparing the tokenization capabilities LaBraM (Jiang et al., 2024) with NEURORVQ. We performed both (*i*) in-distribution and (*ii*) out-of-distribution signal reconstruction comparisons at a per-patch level.

### 5.2.1 IN-DISTRIBUTION EVALUATION

To ensure a fair comparison and to properly evaluate the in-distribution reconstruction capabilities, both architectures (LaBraM and NEURORVQ) were trained on **the same datasets**, with the **same train-validation split**, and **identical training regime**, as described in Section 5.1 (see Appendix E for the learning curves). Figure 4 illustrates the reconstruction of a single EEG signal/patch using the LaBraM and NEURORVQ tokenizers.

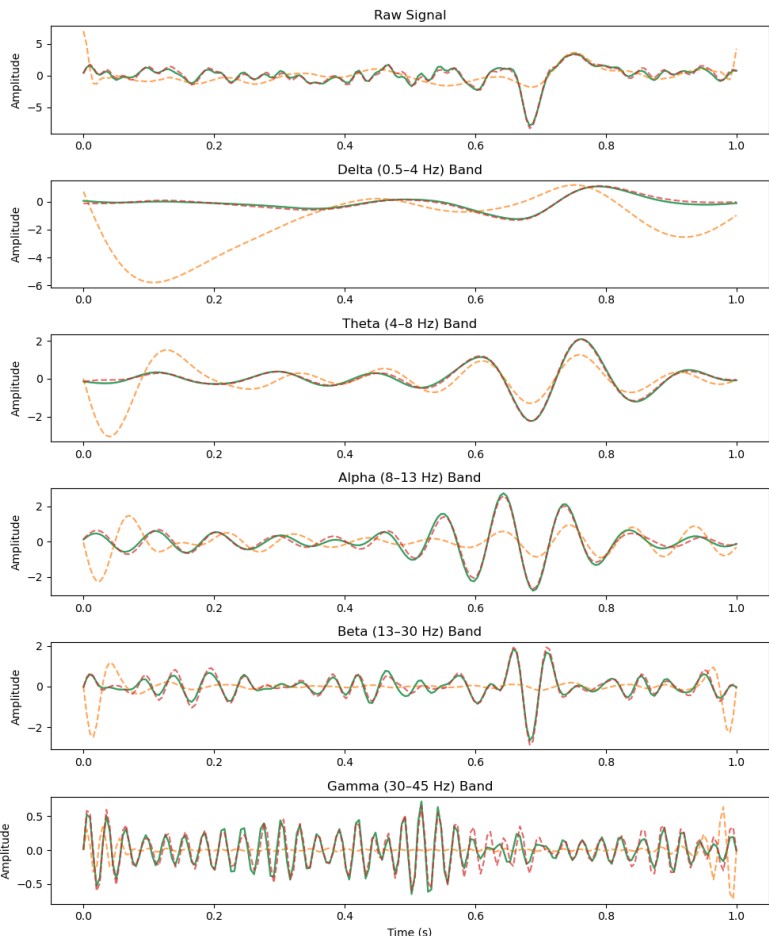

Figure 4: Per-Band Analysis of Reconstructed EEG signal from LaBraM and NEURORVQ codebook-based tokenizers. Green lines denote the input EEG signal, orange the reconstructed EEG signal for LaBraM (ours) and red the reconstructed EEG signal for NEURORVQ.

Table 1 reports the overall reconstruction performance in the validation set across frequency bands. Across all bands, NEURORVQ achieves orders-of-magnitude lower reconstruction error compared to LaBraM, demonstrating that NEURORVQ can produce faithful and robust representation of EEG signals across both broadband and band-specific components.

Table 1: In-distribution validation mean squared error (MSE) reconstruction performance across frequency bands for NEURORVQ and LaBraM tokenizers. Both tokenizers were trained for 100 epochs.

|              | Raw Signal | Delta | Theta | Alpha | Beta  | Gamma |
|--------------|------------|-------|-------|-------|-------|-------|
| LaBraM (ours) | 1.071      | 1.561 | 0.184 | 0.099 | 0.122 | 0.020 |
| NEURORVQ     | **0.016**  | **0.006** | **0.002** | **0.002** | **0.005** | **0.002** |

### 5.2.2 OUT-OF-DISTRIBUTION EVALUATION

We further evaluated the generalization capabilities of both NEURORVQ and LaBraM in reconstructing EEG signals in two datasets unseen during training: a motor dataset High Gamma (Schirrmeister et al., 2017) and a working memory dataset (Pavlov et al., 2022). For LaBraM, we used two variations: (*i*) the version trained with the same datasets as NEURORVQ, highlighted as *ours*; and (*ii*) the authors' pre-trained model, highlighted as *original*. For each dataset, input patches were extracted using the class-related parts of each trial in the dataset.

Tables 2 and 3 show that NEURORVQ consistently outperforms both variations of LaBraM across all frequency bands, demonstrating the robustness and superior generalization of NEURORVQ, even when evaluated on EEG signals from entirely unseen datasets.

Table 2: Out-of-distribution mean squared error (MSE) reconstruction performance on memory task across frequency bands for NEURORVQ and LaBraM tokenizers (original pre-trained and ours).

|              | Raw Signal | Delta | Theta | Alpha | Beta  | Gamma |
|--------------|------------|-------|-------|-------|-------|-------|
| LaBraM (orig.) | 1.985    | 1.093 | 0.216 | 0.191 | 0.211 | 0.054 |
| LaBraM (ours) | 1.880     | 2.757 | 0.361 | 0.352 | 0.161 | 0.031 |
| NEURORVQ     | **0.084**  | **0.046** | **0.006** | **0.009** | **0.022** | **0.010** |

Table 3: Out-of-distribution mean squared error (MSE) reconstruction performance on motor task across frequency bands for NEURORVQ and LaBraM tokenizers (original pre-trained and ours).

|              | Raw Signal | Delta | Theta | Alpha | Beta  | Gamma |
|--------------|------------|-------|-------|-------|-------|-------|
| LaBraM (orig.) | 2.004    | 0.863 | 0.326 | 0.400 | 0.350 | 0.064 |
| LaBraM (ours) | 1.893     | 3.313 | 0.669 | 0.721 | 0.339 | 0.049 |
| NEURORVQ     | **0.090**  | **0.032** | **0.010** | **0.017** | **0.038** | **0.009** |

## 5.3 DOWNSTREAM PERFORMANCE

We evaluated the downstream task performance of fine-tuned NEURORVQ in comparison to other fine-tuned LBMs, namely NeuroGPT (Cui et al., 2024), CBraMod (Wang et al., 2025), LaBraM (Jiang et al., 2024), EEGPT (Wang et al., 2024), BIOT (Yang et al., 2023)[1] and MIRepNet (Liu et al., 2025)[2]. All models were evaluated in four downstream classification tasks (Lee et al., 2025): (*a*) Motor, a four-class movement classification on High Gamma (Schirrmeister et al., 2017) dataset; (*b*) Memory, a binary working memory classification on Pavlov et al. (2022); (*c*) Sleep, a 6-class sleep stage classification on Physionet's Sleep-EDF (Kemp et al., 2000); and (*d*) Eyes, eyes open vs. closed classification on Physionet's Motor (Schalk et al., 2004).These tasks capture a diverse range of BCI paradigms and the datasets were specifically chosen for their minimal spurious artifacts, reducing the likelihood of spurious performance during training (Lee et al., 2025).

We applied the same full fine-tuning procedure to all models, training for 20 epochs since performance plateaued beyond that point. For the sleep task, however, we employed early stopping, as the models

---

[1]BIOT could not be tested on Sleep since the benchmark electrodes are missing from the pre-trained model.
[2]As a motor-related foundation model, MIRepNet was tested only on the motor task

began to overfit before reaching 20 epochs. For evaluation, we used a consistent 10-fold subject-independent cross-validation across all models and reported balanced accuracy for each task. (For downstream performance comparison between codebook-based LBMs pre-trained on the same datasets, see Appendix F).

Table 4: Comparison of classification balanced accuracy between NEURORVQ and other LBMs, finetuned on the same datasets using identical training settings (* denotes early stopping). Bold and underlined values indicate best performance and next-best performance respectively (per task or overall).

| Model | Motor | ERP | Memory | Sleep* | Eyes | Mean |
|---|---|---|---|---|---|---|
| NeuroGPT | 0.682±0.083 | 0.757±0.048 | **0.597±0.029** | 0.674±0.033 | 0.827±0.036 | 0.707±0.046 |
| CBraMod | 0.614±0.104 | 0.777±0.052 | 0.574±0.038 | 0.635±0.041 | 0.839±0.041 | 0.688±0.055 |
| BIOT | 0.443±0.079 | 0.500±0.000 | 0.510±0.018 | – | 0.763±0.049 | – |
| MIRepNet | 0.689±0.086 | – | – | – | – | – |
| LaBraM | 0.630±0.076 | 0.822±0.040 | 0.526±0.026 | 0.652±0.037 | 0.799±0.047 | 0.686±0.045 |
| EEGPT | 0.313±0.035 | 0.668±0.146 | 0.520±0.017 | 0.634±0.044 | 0.797±0.037 | 0.587±0.056 |
| NEURORVQ | **0.700±0.073** | **0.876±0.033** | 0.574±0.027 | **0.728±0.028** | **0.869±0.026** | **0.749±0.037** |

Table 4 shows that, when full-finetuned, NEURORVQ achieves the best or next-best performance in each task, and the best overall performance, measured as the mean accuracy across all tasks. NEURORVQ has 5.9M parameters in the backbone model and approximately 3,000 parameters in the classification heads (see Appendix G). Yet, it achieves higher mean accuracy than NeuroGPT, which has 79.5M parameters in its backbone; and it also outperforms CBraMod, which has similar backbone parameters to NEURORVQ but much larger classification heads (approximately 50M parameters; see Appendix G for a detailed comparison of classification head sizes). This combination of strong generalization and compact size highlights the effectiveness of our NEURORVQ tokenizer.

# 6 DISCUSSION

The NEURORVQ tokenizer demonstrates state-of-the-art reconstruction capabilities on both in-distribution and out-of-distribution signals, as shown in Section 5.2. Importantly, these results cannot be attributed solely to the size of the codebook but also reflect the inductive biases embedded in the tokenizer design together with the specifically designed training process, which are particularly well suited to the characteristics and complex nature of EEG signals. The main contribution of this work is the NEURORVQ codebook, which can be integrated into larger LBM architectures and may enable further performance gains. Further improvements could also come from incorporating training principles inspired by causal reasoning, such as more targeted temporal and spatial masking strategies (Barmpas et al., 2024a). All these directions are beyond the scope of this work and could be promising avenues for future research.

# 7 CONCLUSION

In this work, we introduce NEURORVQ, an LBM that leverages a Residual Vector Quantization (RVQ) codebook-based tokenizer. Our main contribution is the design and proper training of the NEURORVQ tokenizer, which captures low- and high-frequency components of EEG signals, achieves state-of-the-art reconstruction performance and facilitates the development of more effective LBMs. We further showed that NEURORVQ outperforms existing pre-trained LBMs on various downstream tasks. In summary, NEURORVQ advances the development of efficient LBMs by providing a high-fidelity tokenizer. The principles underlying NEURORVQ's tokenizer can be extended to other biosignals (see Appendix H), paving the way for broader applications of foundation models in the biosignal space.

## LLM DECLARATION

Portions of this manuscript were edited for clarity and grammar with the assistance of a large language model. All scientific content, results and conclusions are the authors' own.

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

## A  SINE-COSINE PHASE REPRESENTATION

We describe the benefits of using a sine-cosine representation of the Fourier phase $\phi$ in the loss function, instead of the raw phase values.

LaBraM (Jiang et al., 2024) computes phase loss directly as the squared error between predicted $\hat{\phi}$ and true $\phi$:

$$\mathcal{L}_\phi = ||\hat{\phi} - \phi||^2. \tag{8}$$

This formulation, however, is discontinuous at the boundaries of the phase domain $[-\pi, \pi]$. For instance, with $\hat{\phi} = \pi - \epsilon$ and $\phi = -\pi + \epsilon$ for small $\epsilon > 0$,

$$\mathcal{L}_\phi = ||\hat{\phi} - \phi||^2 = (2\pi - 2\epsilon)^2. \tag{9}$$

The loss approaches $4\pi^2$, even though the two angles are nearly identical. Thus, small phase shifts near the $\pm\pi$ boundary (e.g., $2°$ between $179°$ and $-179°$) result in large discontinuous jumps in the loss, leading to unstable gradients and poor convergence.

To address this issue, we map phase $\phi$ to its sine and cosine components, a point on the unit circle:

$$r(\phi) = (\sin(\phi), \cos(\phi)) \in \mathbb{R}^2, \tag{10}$$

The corresponding loss is:[3]

$$\mathcal{L}_{r(\phi)} = ||\sin(\hat{\phi}) - \sin(\phi)||^2 + ||\cos(\hat{\phi}) - \cos(\phi)||^2 = 2 - 2\cos(\hat{\phi} - \phi) \tag{11}$$

Our formulation is continuous across the $\pm\pi$ boundary. Revisiting the earlier example with $\hat{\phi} = \pi - \epsilon$ and $\phi = -\pi + \epsilon$,

$$\mathcal{L}_{r(\phi)} = 2 - 2\cos(2\epsilon) \approx 4\epsilon^2 \tag{12}$$

which smoothly approaches 0 as $\epsilon \to 0$.

More generally, for any two phases $\phi_1$ and $\phi_2$:

$$\mathcal{L}_{r(\phi)} = 2 - 2\cos(\phi_1 - \phi_2) = 4\sin^2\left(\frac{\phi_1 - \phi_2}{2}\right). \tag{13}$$

> $\mathcal{L}_{r(\phi)}$ is the squared chord length on the unit circle between points at angles $\phi_1$ and $\phi_2$.

## B  MODEL CONFIGURATION & HYPERPARAMETER SETTINGS

This section details: (*i*) the configuration of NEURORVQ's multi-scale temporal encoder (§B.1), and (*ii*) the hyperparameter settings used for training the tokenizer and foundation model, as well as fine-tuning the latter on downstream tasks (§B.2).

### B.1  MULTI-SCALE TEMPORAL ENCODER ARCHITECTURE

Table 5: Configuration of the multi-scale temporal encoder, used identically in both the tokenizer and the foundation model. Here, $x \to y$ indicates values for the first ($x$) and second ($y$) sequence.

| Branch № | Filters | Kernel | Padding | Pooling |
|---|---|---|---|---|
| 1 | $8 \to 8$ | $(1, 21) \to (1, 9)$ | $(0, 10) \to (0, 4)$ | $(1, 2) \to (1, 4)$ |
| 2 | $8 \to 8$ | $(1, 15) \to (1, 7)$ | $(0, 7) \to (0, 3)$ | $(1, 2) \to (1, 4)$ |
| 3 | $8 \to 8$ | $(1, 9) \to (1, 5)$ | $(0, 4) \to (0, 2)$ | $(1, 2) \to (1, 4)$ |
| 4 | $8 \to 8$ | $(1, 5) \to (1, 3)$ | $(0, 2) \to (0, 1)$ | $(1, 2) \to (1, 4)$ |

Our temporal encoder comprises four temporal branches. Each branch applies the following sequence of operations twice in succession (Figure 1c): an 1-D convolution, followed by group normalization with $N = 4$ groups and $C = 8$ channels, GELU activation, and pooling. Table 5 lists the filter counts, kernel sizes, padding, and pooling parameters for the two sequences, where $x \to y$ denotes the configuration value(s) used in the first and second sequence, respectively.

---

[3]Although the NEURORVQ architecture uses the $\mathcal{L}_{\text{unit-loss}}$ as described in the manuscript, we use $\mathcal{L}_{r(\phi)}$ here to illustrate the effectiveness of the sine-cosine representation.

## B.2 HYPERPARAMETER SETTINGS

Table 6: Hyperparameters

| Hyperparameter | Tokenizer | Pre-training FM | Finetuning |
|---|---|---|---|
| Temporal Inception Branches $S$ | 4 | 4 | 4 |
| RVQ Codebook $\mathcal{R}$ | 4 | 4 | 4 |
| Single VQ Codebook $\mathcal{V}_i$ | 8 | 8 | 8 |
| Batch size | 256 | 64 | 64 |
| Learning rate scheduler | Cosine | Cosine | Linear |
| Base learning rate | 5e-5 | 5e-4 | 5e-4 |
| Min learning rate | 1e-5 | 1e-5 | - |
| Warmup lr start-end factors | - | - | (0.1,1) |
| Lr start-end factors | - | - | (1,0.1) |
| Total epochs | 100 | 50 | 20 |
| Warmup epochs | 10 | 5 | 4 |
| Optimizer | AdamW | AdamW | AdamW |
| Weight decay | 1e-4 | 0.05 | 0.01 |
| Adam $\beta$ | (0.9, 0.999) | (0.9, 0.999) | (0.9, 0.999) |
| Layer lr decay | - | - | 0.975 |
| Gradient clipping | 3 | - | - |
| Layer scale init | 0.001 | 0.001 | - |
| Encoder depth | 12 | 12 | 12 |
| Decoder depth | 3 | - | - |
| Hidden dimension | 200 | 200 | 200 |
| No. Attention heads | 10 | 10 | 10 |
| MLP hidden dimension | 800 | 800 | 800 |

Table 6 summarises the hyperparameters used for training the tokenizer and for pre-training and fine-tuning the foundation model. Certain choices (e.g., batch size) were constrained by hardware, while others were guided by ablation studies (e.g., using 8 codebooks per RVQ yielded the lowest reconstruction error; see §C).

## C NUMBER OF CODEBOOKS PER RVQ

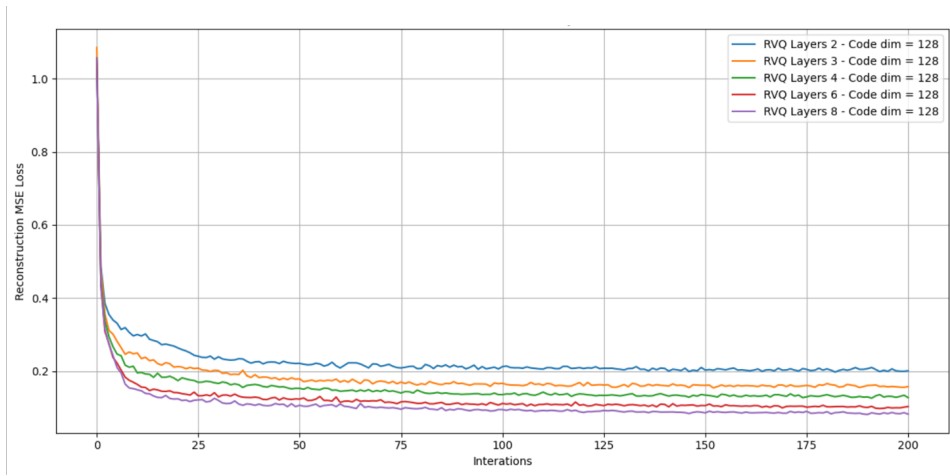

Figure 5: Validation MSE reconstruction loss for the NEURORVQ tokenizer with different numbers of codebooks (layers) per RVQ.

We conducted an ablation study to determine the best number of codebooks (layers) per RVQ. The tokenizer was trained using our self-collected dataset (see §D), varying the number of layers from 2 to 8. Figure 5 shows that using eight layers $\mathcal{V}_i \in \mathbb{R}^{8192 \times 128}$ achieve the best reconstruction performance on the validation set.

# D  DATASETS

Table 7: Datasets used for pre-training NEURORVQ, with BCI paradigm and number of EEG recording channels.

| Datasets | BCI paradigm | # channels |
|---|---|---|
| BCI Competition IV-1 (Blankertz et al., 2007) | Motor | 64 |
| Grasp and Lift (Luciw et al., 2014) | Motor | 32 |
| Physionet MI (Schalk et al., 2004) | Motor | 64 |
| bi2015a (Korczowski et al., 2019) | ERP | 32 |
| Inria BCI Challenge (Margaux et al., 2012) | ERP | 56 |
| Trujillo (2020) | II-RB | 64 |
| Trujillo et al. (2017) | Resting | 64 |
| SPIS Resting (Torkamani-Azar et al., 2019) | Resting | 64 |
| TUAR (Buckwalter et al., 2021) | Artifacts (e.g., eye movement) | 31 |
| Siena Scalp (Detti et al., 2020) | Epilepsy | 31 |
| TUEP (Veloso et al., 2017) | Epilepsy | 31 |
| TUSZ (Shah et al., 2018) | Seizure | 31 |
| Self-Collected Dataset | Motor | 29 |

We use a collection of publicly available EEG datasets to pre-train NEURORVQ. Table 7 summarizes the datasets, their associated BCI paradigms, and the number of EEG recording channels. The collection spans multiple experimental settings, including motor imagery, event-related potentials (ERP), resting-state activity, information-integration and rule-based cognitive categorization (II-RB), and clinical data for epilepsy and seizures.

We additionally include a proprietary dataset comprising approximately 235 hours of motor imagery data recorded with 29 channels. Participants in this data collection were instructed as follows:

> "We will record your brain waves while you engage in simple tasks. These include real and imagined movements of the extremities, and a game which uses movements as controls. The collected data will be used to train algorithms that can be used for many things, such as facilitating navigation in virtual environments, seamless user interaction with objects in computer gaming or even rehabilitation training of post-stroke patients towards regaining the ability to control their limbs. The project received approval from the Research Governance and Integrity Team. If you want to help out our research team you can register to be a participant in our experiment! There is a small gift of £30 to every participant."

## E    NEURORVQ VS. LABRAM: RECONSTRUCTION PERFORMANCE

We train NEURORVQ and LaBraM tokenizers on the same datasets with the same train–validation split and an identical training regime (including preprocessing and training steps), as described in §5.1. Figure 6 shows the validation learning curves for both models. NEURORVQ achieves nearly two orders of magnitude lower reconstruction MSE compared to LaBraM.

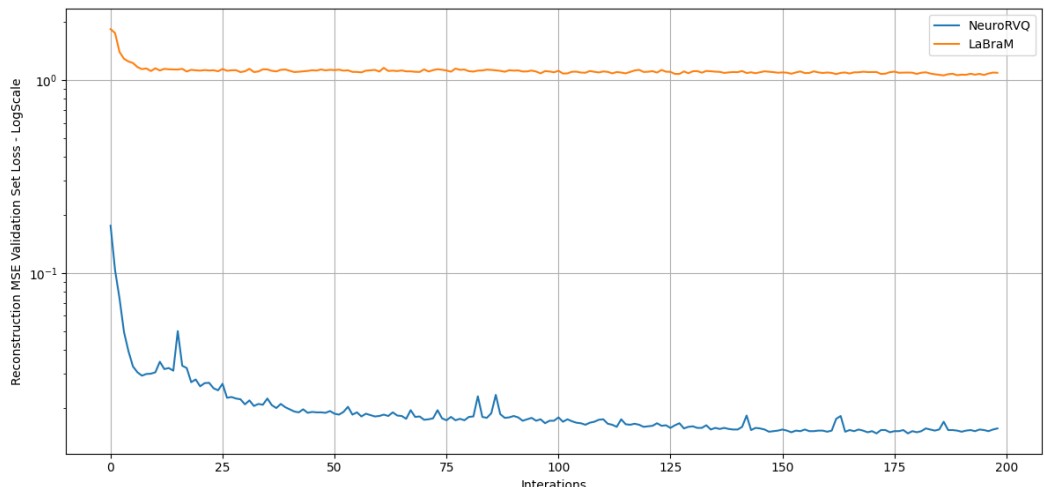

Figure 6: Validation learning curves showing reconstruction MSE for NEURORVQ and LaBraM tokenizers, trained under identical regimes.

## F    NEURORVQ VS. LABRAM: DOWNSTREAM TASK PERFORMANCE

We compare the downstream task performance of NEURORVQ with LaBraM when both models are pre-trained **and** fine-tuned on the same datasets and with an identical training regime. We use 10-fold subject-independent cross-validation to evaluate models on the four classification benchmarks described in §5.3. Table 8 shows that, when all else is equal, NEURORVQ achieves 1%–13% higher balanced accuracy than LaBraM on all tasks, demonstrating the effectiveness of our tokenizer in training codebook-based models.

Table 8: Comparison of classification balanced accuracy between NEURORVQ and LaBraM, both pre-trained and finetuned on the same datasets using identical settings. (* denotes early stopping)

| Model | Motor | Memory | Sleep* | Eyes |
|---|---|---|---|---|
| LaBraM (ours) | 0.570 | 0.565 | 0.715 | 0.805 |
| NEURORVQ | 0.700 | 0.573 | 0.728 | 0.869 |

# G  Classification Heads: Architecture & Size

Earlier, in Table 4, we compared the downstream classification performance of NEURORVQ against five state-of-the-art models: CBraMod, BIOT, EEGPT, NeuroGPT and LaBraM. Here, we report the number of parameters in the task-specific classification head for each model. CBraMod and NeuroGPT employ a 3-layer MLP as their classification heads, according to their original implementations, whereas the others use a single layer. The head sizes vary with the number of target classes (Motor has 4, Memory 2, Sleep 6 and Eyes 2); and for CBraMod, also with the number of EEG channels. Table 9 show that NEURORVQ has relatively low number of parameters compared to other models, both in the backbone and in the classification heads.

Table 9: Parameter counts for model backbones and task-specific classification heads.

| Model | Backbone | Motor Head | Memory Head | Sleep Head | Eyes Head |
|-------|----------|------------|-------------|------------|-----------|
| CBraMod | 4.9M | 50,081,804 | 40,481,201 | 73,207,406 | 41,121,201 |
| NeuroGPT | 79.5M | 270,756 | 270,657 | 270,822 | 270,657 |
| BIOT | 3.2M | 1,028 | 257 | - | 257 |
| LaBraM | 5.8M | 804 | 201 | 1,206 | 201 |
| EEGPT | 25.7M | 260 | 65 | 390 | 65 |
| NEURORVQ | 5.9M | 3,204 | 801 | 4,806 | 801 |

# H  Tokenization of Biosignal Modalities

The principles underlying NEURORVQ's tokenizer apply to other biosignals, enabling broader applications of foundation models in the biosignal domain. In this section, we present signal reconstruction results on two additional modalities: electromyography (EMG) and electrocardiogram (ECG).

For ECG, we used the PTB-XL dataset (Wagner et al., 2022), resampled at 200Hz. The NEURORVQ tokenizer was trained using the settings detailed in Appendix B. Training was conducted for 100 epochs with $S = 4$ temporal branches and 4 $\mathcal{R}$ RVQ codebooks, each consisting of 8 single codebooks $\mathcal{V}i \in \mathbb{R}^{8192 \times 128}$, with $\lambda_{circle} = 0.4$. Figure 7 shows signal reconstruction for an ECG signal, demonstrating excellent fidelity.

For EMG, we used the emg2pose dataset (Salter et al., 2024), resampled at 1000Hz. The NEURORVQ tokenizer used the same configuration as above, except that each of the 4 $\mathcal{R}$ RVQ codebooks contained 16 single codebooks $\mathcal{V}_i \in \mathbb{R}^{8192 \times 128}$ and the kernel sizes appropriately adapted as detailed in Table 10. Figure 8 shows signal reconstruction for an EMG signal, showing high accuracy in preserving its original features.

Table 10: Configuration of the multi-scale temporal encoder used in the EMG tokenizer. Here, $x \to y$ indicates values for the first ($x$) and second ($y$) sequence.

| Branch № | Filters | Kernel | Padding | Pooling |
|----------|---------|--------|---------|---------|
| 1 | $8 \to 8$ | $(1, 51) \to (1, 25)$ | $(0, 25) \to (0, 12)$ | $(1, 2) \to (1, 4)$ |
| 2 | $8 \to 8$ | $(1, 17) \to (1, 9)$ | $(0, \ 8) \to (0, 4)$ | $(1, 2) \to (1, 4)$ |
| 3 | $8 \to 8$ | $(1, \ 8) \to (1, 4)$ | $(0, \ 4) \to (0, 2)$ | $(1, 2) \to (1, 4)$ |
| 4 | $8 \to 8$ | $(1, \ 5) \to (1, 3)$ | $(0, \ 2) \to (0, 1)$ | $(1, 2) \to (1, 4)$ |

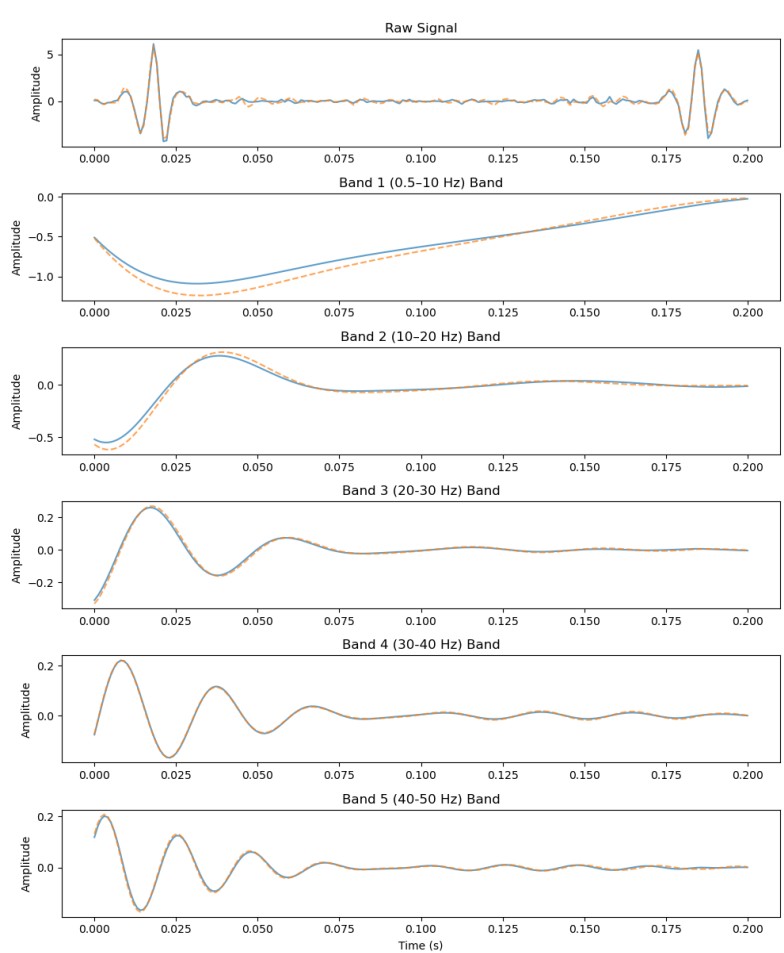

Figure 7: Per-Band Analysis of Reconstructed ECG signal from NEURORVQ codebook-based tokenizer. Blue lines denote the input EMG signal and orange the reconstructed ECG signal for NEURORVQ.

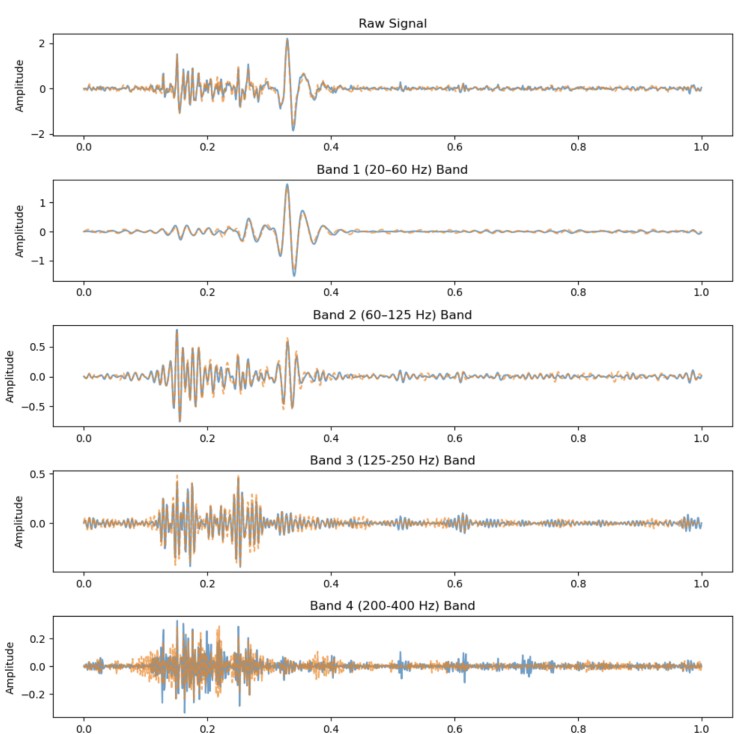

Figure 8: Per-Band Analysis of Reconstructed EMG signal from NEURORVQ codebook-based tokenizer. Blue lines denote the input EMG signal and orange the reconstructed EMG signal for NEURORVQ.

