# OpenReview forum: "NeuroRVQ: Multi-Scale EEG Tokenization for Generative Large Brainwave Models"
_ICLR.cc/2026/Conference — Submitted to ICLR 2026_

### Official Review · Reviewer_maKs · 2025-10-27

**Soundness:** 2
**Presentation:** 3
**Contribution:** 3
**Rating:** 4
**Confidence:** 4

**Summary:**

The paper introduces NeuroRVQ, a scalable Large Brainwave Model (LBM) for EEG signal modeling that addresses limitations in existing signal tokenization methods. It employs a codebook-based tokenizer integrating (i) multi-scale feature extraction, (ii) hierarchical residual vector quantization, and (iii) a phase- and amplitude-aware loss for efficient training. This design enables accurate reconstruction across all frequency bands and efficient EEG compression, outperforming existing approaches in reconstruction quality and downstream tasks.

**Strengths:**

- Although using frequency-domain information across EEG bands is common in existing brainwave foundation models, this paper’s approach effectively leverages such domain knowledge grounded in neuroscience principles, leading to more biologically informed EEG modeling.

- The paper offers clear and insightful analysis on the role of the EEG tokenizer, and proposes a well-designed hierarchical improvement strategy that enhances existing tokenization methods. The methodology is technically sound and clearly presented.

- The case analysis in Figure 4 vividly and intuitively demonstrates the performance advantages of the proposed codebook-based tokenizer, helping readers grasp the model’s effectiveness.

**Weaknesses:**

- The authors report training the tokenizer for 100 epochs and the foundation model for 50 epochs. For large models, this is unusually high — typical large-scale training often involves vast data with fewer epochs (e.g., 1–2). Such a high number may suggest insufficient training data or diversity, or risk overfitting.

- Although the paper lists the datasets used, the number of subjects in the training and evaluation sets is not clearly specified. This metric is crucial for assessing generalization, as limited subject diversity can restrict the universality of conclusions.

- The baselines in the four classification tasks are not comprehensive enough. For example, Table 4 includes models up to 79.5M parameters, but omits larger baselines such as Brant (≈500M), which could weaken the strength of the empirical comparisons.

- The absence of released code is a reproducibility concern, especially for a work proposing a new large-scale EEG foundation model.

**Questions:**

1. The authors mention training the tokenizer for 100 epochs and the foundation model for 50 epochs. Was this choice due to limited data quantity or diversity, or could it risk overfitting? This point should be more thoroughly discussed.

2. How were the hyperparameters (e.g., S = 4, λ = 0.4) selected? Were hyperparameter sensitivity analyses conducted to justify these values?

3. Please report the number of subjects in the training and evaluation datasets, as this is essential for assessing cross-subject generalization.

4. In the in-distribution and out-of-distribution experiments, the tokenizer is compared only with LaBraM. Could the limited number of baselines undermine the strength of the conclusions?

5. The authors are encouraged to include missing baselines (such as the series work of Brant [1] and Brant-2 [2]), or at least explain why these works are not suitable for comparison, and discuss them in the related work section to reflect completeness and fairness.

6. As this is a large-scale EEG model, have the authors considered testing the model and baselines under a zero-shot setting? This would better demonstrate the generalization capabilities of the proposed approach.

7. It is recommended that the authors release their model code to promote reproducibility and community engagement.

> [1] Brant: foundation model for intracranial neural signal. (https://papers.neurips.cc/paper_files/paper/2023/file/535915d26859036410b0533804cee788-Paper-Conference.pdf)
>
> [2] Brant-2: Foundation Model for Brain Signals. (https://www.arxiv.org/abs/2402.10251v4)

---

> ### Author Response · Authors · 2025-11-19
> **Response to Reviewer maKs**
>
> We would like to thank the reviewer for their thoughtful review and valuable feedback.
>
> **Responses to Reviewer's Questions**
>
> **Q1.**
> As stated in the manuscript, the tokenizer was trained for 100 epochs and the foundation model for 50 epochs. These values align with the training of other SOTA codebook-based models (e.g., LaBraM). In the validation learning curves (example shown in Figure 5), we did not witness signs of overfitting.
>
> **Q2.**
> The value of S=4 was selected heuristically based on neuroscientific findings. Essentially, it was the smallest number that still enables effective disentanglement of the canonical EEG frequency bands (e.g., alpha, beta, etc). Increasing this number would allow the model to capture even finer details but that would increase the model's complexity with diminishing returns. Based on our initial results, the S=4 was a good compromise of quality and model complexity. The value of l=0.4 for regularization term is a commonly reasonable choice in similar settings. We can add this explanation to our revised manuscript.
>
> **Q3.**
> We thank the reviewer for this comment. We will add these numbers in the revised manuscript. For the evaluation datasets, we used the datasets based on the [*] benchmark: High Gamma (14 subjects),  Pavlov 2022 (65 subjects), OPENBMI-ERP (54 subjects), SLEEP-EDF (78 subjects) and Physionet (103 subjects).
>
> During the review period, we have also added results for the last task of this benchmark (ERP detection), which further supports the generalization and practical value of our approach.
>
> | Model     | ERP               |
> |-----------|-------------------|
> | NeuroGPT  | 0.757 ± 0.048     |
> | CBraMod   | 0.777 ± 0.052     |
> | BIOT      | 0.500 ± 0.000     |
> | LaBraM    | 0.822 ± 0.040     |
> | EEGPT     | 0.668 ± 0.146     |
> | **NeuroRVQ** | **0.876 ± 0.033** |
>
> [*] Na Lee et al. Assessing the Capabilities of Large Brainwave Foundation Models. In 2025 IEEE 35th International Workshop on Machine Learning for Signal Processing (MLSP) and ICLR2025 Workshop on Spurious Correlation and Shortcut Learning: Foundations and Solutions, 2025.
>
> **Q4.**
> In both the in-distribution and out-of-distribution experiments, we compare NeuroRVQ with the current SOTA codebook-based tokenizer. Importantly, our comparison is not limited to the open-source pretrained weights of this tokenizer. We also trained our own version from scratch using the same datasets as NeuroRVQ. This ensures a fair and rigorous evaluation and highlights the strength of our methodology, demonstrating that NeuroRVQ consistently outperforms the current SOTA codebook-based approach. The only other codebook-based tokenizer is BrainOmni [1] that was published after the submission. We can observe from Figure 4 of BrainOmni that they do not reconstruct high frequency.
>
> [1] Xiao Q, Cui Z, Zhang C, et al. BrainOmni: A Brain Foundation Model for Unified EEG and MEG Signals[J]. arXiv preprint arXiv:2505.18185, 2025.
>
> **Q5.**
> The main reason we did not include Brant is due to its pretraining in iEEG data, which makes it inherently hard to properly evaluate them in the downstream tasks datasets of the benchmarks we chose. But as the reviewer suggested we will include them in related work for completeness in the revised manuscript.
>
> **Q6.**
> While few-shot and even zero-shot model evaluation would indeed be interesting, unfortunately, neither NeuroRVQ nor comparable models are currently capable of such  task generalization. Like other foundation models, NeuroRVQ is trained with masked patch prediction to reconstruct missing EEG segments. This training objective is fundamentally focused on signal reconstruction and is not aligned with task-specific generalization. Consequently, while NeuroRVQ serves as a strong model prior and excels at reconstructing EEG signals, it cannot directly generalize to new downstream tasks without additional supervision or fine-tuning. For this reason, supervised downstream task performance is used as the evaluation metric across all related works. As shown in Table 4, our model consistently outperforms competing approaches on these tasks.
>
> **Q7.**
> The code and pre-trained weights will become available upon acceptance.
>
> **Conclusion.**
> We sincerely thank the reviewer for their valuable comments, which have helped us to improve our work. We hope the reviewer recognizes the potential impact of this work as a meaningful and timely contribution to the ICLR community.

---

> > ### Comment · Reviewer_maKs · 2025-11-24
> >
> > The authors’ response has addressed most of the issues, and I have therefore raised my score. I hope the authors will faithfully incorporate the revisions into the new version of the paper.

---

> > > ### Author Response · Authors · 2025-11-26
> > > **Response to reviewer maKs**
> > >
> > > We sincerely thank the reviewer for their thoughtful feedback and for raising their score. The revisions will be incorporated in the new version of the paper. If there are any additional questions or points requiring clarification, we would be happy to address them.

---

> > > > ### Author Response · Authors · 2025-12-01
> > > > **Final Comment to Reviewer maKs**
> > > >
> > > > We thank the reviewer for their feedback and for **substantially raising their score**. We have incorporated all changes in our revised manuscript.
> > > >
> > > > **Downstream Task Performance**
> > > > In our updated manuscript, we have also added a newly released Motor EEG Foundation Model, MIRepNet. The final performance in downstream tasks can be summarised as follows:
> > > >
> > > > | Model       | Motor        | ERP          | Memory        | Sleep      | Eyes         | Mean         |
> > > > |------------|-------------|-------------|--------------|-------------|-------------|-------------|
> > > > | NeuroGPT    | 0.682±0.083  | 0.757±0.048 | **0.597±0.029** | 0.674±0.033 | 0.827±0.036 | 0.707±0.046 |
> > > > | CBraMod     | 0.614±0.104  | 0.777±0.052 | 0.574±0.038 | 0.635±0.041 | 0.839±0.041 | 0.688±0.055 |
> > > > | BIOT        | 0.443±0.079  | 0.500±0.000 | 0.510±0.018 | --          | 0.763±0.049 | --          |
> > > > | MIRepNet    | 0.689±0.086  | --          | --          | --          | --          | --          |
> > > > | LaBraM      | 0.630±0.076  | 0.822±0.040 | 0.526±0.026 | 0.652±0.037 | 0.799±0.047 | 0.686±0.045 |
> > > > | EEGPT       | 0.313±0.035  | 0.668±0.146 | 0.520±0.017 | 0.634±0.044 | 0.797±0.037 | 0.587±0.056 |
> > > > | **NeuroRVQ** | **0.700±0.073** | **0.876±0.033** | 0.574±0.027 | **0.728±0.028** | **0.869±0.026** | **0.749±0.037** |
> > > >
> > > > **Open-Source Code**
> > > > In response to the reviewer’s comment, we have prepared a repo to provide our models open-source (upon acceptance), with a demo functionality for users to see the tokenization capabilities.

---

### Official Review · Reviewer_n4Y3 · 2025-10-30

**Soundness:** 2
**Presentation:** 2
**Contribution:** 2
**Rating:** 2
**Confidence:** 4

**Summary:**

This paper introduces NeuroRVQ, a scalable Large Brainwave Model (LBM) architecture for EEG data, which features multi-scale temporal convolutions, hierarchical residual vector quantization (RVQ) codebooks, and a training loss that jointly reconstructs phase and amplitude information using a sine-cosine representation. Extensive experiments demonstrate superior reconstruction error and downstream classification performance compared to prior codebook and transformer-based LBMs.

**Strengths:**

The manuscript is clearly written and well-structured. The extensive use of tables and figures effectively presents detailed results, ablation studies, and interpretability analyses, which greatly clarify the authors' model choices. The supplemental materials are thorough and provide valuable additional support.

**Weaknesses:**

1. Applying RVQ and phase-aware loss to EEG modeling is not novel; see more details in [1,2]. For example, BrainOmni replaces the original VQ with RVQ to separate different source components, and combines both time and frequency domain loss to guide pretraining.

2. The datasets evaluated in this work (Table 4) appear to have little overlap with the datasets evaluated by the baseline models [3-7], which poses a significant challenge to head-to-head model comparisons.

3. The multi-scale feature extraction module does not appear to integrate information from different scales into a single token (Line 740). While this operation does not affect the model size, it significantly increases computation time. Furthermore, this operation constitutes an implicit model ensemble, leading to unfair comparisons between models.

**References**:

[1] Xiao Q, Cui Z, Zhang C, et al. BrainOmni: A Brain Foundation Model for Unified EEG and MEG Signals[J]. arXiv preprint arXiv:2505.18185, 2025.

[2] Carzaniga F S, Hoppeler G T, Hersche M, et al. The Case for Cleaner Biosignals: High-fidelity Neural Compressor Enables Transfer from Cleaner iEEG to Noisier EEG[J]. arXiv preprint arXiv:2502.17462, 2025.

[3] Wang J, Zhao S, Luo Z, et al. Cbramod: A criss-cross brain foundation model for eeg decoding[J]. arXiv preprint arXiv:2412.07236, 2024.

[4] Jiang W B, Zhao L M, Lu B L. Large brain model for learning generic representations with tremendous EEG data in BCI[J]. arXiv preprint arXiv:2405.18765, 2024.

[5] Cui W, Jeong W, Thölke P, et al. Neuro-gpt: Towards a foundation model for eeg[C]//2024 IEEE International Symposium on Biomedical Imaging (ISBI). IEEE, 2024: 1-5.

[6] Yang C, Westover M, Sun J. Biot: Biosignal transformer for cross-data learning in the wild[J]. Advances in Neural Information Processing Systems, 2023, 36: 78240-78260.

[7] Wang G, Liu W, He Y, et al. Eegpt: Pretrained transformer for universal and reliable representation of eeg signals[J]. Advances in Neural Information Processing Systems, 2024, 37: 39249-39280.

**Questions:**

See weaknesses.

---

> ### Author Response · Authors · 2025-11-19
> **Response to Reviewer n4Y3**
>
> We would like to thank the reviewer for their thoughtful review and valuable feedback.
>
> **Responses to Reviewer's Questions**
>
> **Q1.**
> Our work solves an open problem in BCI foundation models: high-fidelity EEG tokenization. Indeed, our solution puts together well-established components, but  their integration is non-trivial; and it also introduces new ones (e.g., our loss function). Together, they make a state-of-the-art EEG tokenizer.
>
> While these two works, BrainOmni and BrainCodec, both use RVQ codebook, they exhibit significant limitations in EEG reconstruction fidelity. BrainOmni [1] uses an RVQ codebook to separate different source components, while NeuroRVQ utilizes RVQ codebooks to capture signal components. BrainCodec [2] is a direct application of EnCodec to the EEG / iEEG domains and relies on a single RVQ codebook and discriminator. Unlike these works, NeuroRVQ employs multiple RVQ codebooks per frequency band to capture fine-grained signal elements.
>
> A qualitative and quantitative comparison further illustrates this difference. As shown in Figure 4 of BrainOmni and Figures A9 and A16 of BrainCodec, these models do not recover high-frequency activity and fine-scale structural details. In contrast, our results in Figure 4 and Tables 1–3 demonstrate substantially improved reconstruction fidelity, addressing this known limitation.
> Note that this improvement arises not simply from the use of RVQ itself, but from a carefully designed training loss.
>
> Our training objective, derived from well-established signal-processing principles, uses amplitude log-loss, sinusoidal (sine/cosine) components and a unit-circle penalty that are, to our knowledge, introduced for the first time in the context of EEG signal reconstruction. These loss components are fundamentally different from those in frameworks such as EnCodec, which rely on discriminator and spectrogram losses (not present in our work) rather than signal motivated constraints.
>
> (BrainOmni was not part of our paper since during the submission deadline it was published to a journal/conference yet.)
>
> **Q2.**
> The reviewer's comment echoes a well-known issue in the BCI Foundation Model research community: the lack of benchmarks. Even the works that the reviewer cited do always not report results on the same datasets. CBraMod uses FACED, SEED and PhysioNet-MI while LaBraM uses TUAB and TUEV.
>
> Our primary objective is to develop a high-fidelity tokenizer capable of accurate signal reconstruction. To this end, we adopt the state-of-the-art codebook-based tokenizer, LaBraM, and conduct a series of experiments comparing both the original open-source model and a version trained from scratch on the same datasets as NeuroRVQ, enabling a fair head-to-head evaluation, as shown in Tables 1-3.
>
> In addition, as outlined in Section 5.3, we adopt a recently published EEG benchmark [*], which is motivated by causal reasoning and the removal of task-discriminative artifacts. To ensure fairness, our pre-training dataset does not include any of the benchmark data, whereas some competing baselines do. During the review period, we have also added results for the last task of this benchmark (ERP detection task), which further supports the generalization and practical value of our approach.
>
> | Model     | ERP               |
> |-----------|-------------------|
> | NeuroGPT  | 0.757 ± 0.048     |
> | CBraMod   | 0.777 ± 0.052     |
> | BIOT      | 0.500 ± 0.000     |
> | LaBraM    | 0.822 ± 0.040     |
> | EEGPT     | 0.668 ± 0.146     |
> | **NeuroRVQ** | **0.876 ± 0.033** |
>
> [*] Na Lee et al. Assessing the Capabilities of Large Brainwave Foundation Models. In 2025 IEEE 35th International Workshop on Machine Learning for Signal Processing (MLSP) and ICLR2025 Workshop on Spurious Correlation and Shortcut Learning: Foundations and Solutions, 2025.
>
> **Q3.**
> We would like to clarify our design. Each EEG patch is processed through parallel convolutional branches to effectively disentangle the underlying frequency components. This structure is not an ensemble, but rather an Inception-style module that extracts multi-scale temporal features.  Since each branch is trainable, it does affect the tokenizer's model size. Importantly, each branch is paired with a distinct RVQ codebook, a necessary design choice that enables the model to learn and preserve the unique characteristics of each frequency band.
>
> During reconstruction, the tokens from all branches are combined to generate the final signal. During foundation-model pretraining, each masked patch predicts tokens across all branches, reflecting band-specific frequency dynamics.
>
> This multi-band token approach mechanism contributes directly to the accurate signal reconstruction (Tables 1-3) and the strong downstream task performance (Table 4).
>
> (In Line 740 we report the parameters associated with the different multi-scale branches.)

---

> > ### Author Response · Authors · 2025-11-19
> > **Response to Reviewer n4Y3 (2)**
> >
> > **Conclusion**
> >
> > We sincerely thank the reviewer for their valuable comments, which have helped us to improve our work. We hope the reviewer, familiar with works like LaBraM and BrainOmni, recognizes the potential impact of this work and the great benefits that EEG tokenization will bring to the BCI Foundation Model research community.

---

> > > ### Author Response · Authors · 2025-11-26
> > > **Follow-up to reviewer n4Y3**
> > >
> > > We would like to kindly follow up regarding our rebuttal response, as we have not yet received further comments from the reviewer. If there are any additional questions or points requiring clarification, we would be happy to address them.

---

> > > > ### Author Response · Authors · 2025-12-01
> > > > **Final Comment to Reviewer n4Y3**
> > > >
> > > > We thank the reviewer for their feedback. We hope that our responses satisfactorily address the reviewers’ questions and concerns.
> > > >
> > > > **Downstream Task Performance**
> > > > In our updated manuscript, we have also added a newly released Motor EEG Foundation Model, MIRepNet. The final performance in downstream tasks can be summarised as follows:
> > > >
> > > > | Model       | Motor        | ERP          | Memory        | Sleep      | Eyes         | Mean         |
> > > > |------------|-------------|-------------|--------------|-------------|-------------|-------------|
> > > > | NeuroGPT    | 0.682±0.083  | 0.757±0.048 | **0.597±0.029** | 0.674±0.033 | 0.827±0.036 | 0.707±0.046 |
> > > > | CBraMod     | 0.614±0.104  | 0.777±0.052 | 0.574±0.038 | 0.635±0.041 | 0.839±0.041 | 0.688±0.055 |
> > > > | BIOT        | 0.443±0.079  | 0.500±0.000 | 0.510±0.018 | --          | 0.763±0.049 | --          |
> > > > | MIRepNet    | 0.689±0.086  | --          | --          | --          | --          | --          |
> > > > | LaBraM      | 0.630±0.076  | 0.822±0.040 | 0.526±0.026 | 0.652±0.037 | 0.799±0.047 | 0.686±0.045 |
> > > > | EEGPT       | 0.313±0.035  | 0.668±0.146 | 0.520±0.017 | 0.634±0.044 | 0.797±0.037 | 0.587±0.056 |
> > > > | **NeuroRVQ** | **0.700±0.073** | **0.876±0.033** | 0.574±0.027 | **0.728±0.028** | **0.869±0.026** | **0.749±0.037** |

---

### Official Review · Reviewer_gbAW · 2025-10-31

**Soundness:** 2
**Presentation:** 3
**Contribution:** 2
**Rating:** 2
**Confidence:** 5

**Summary:**

This paper presents NeuroRVQ, a large brainwave model (LBM) for EEG signals that focuses on a multi-scale tokenizer using temporal convolutions with varying kernel sizes for frequency band extraction, hierarchical residual vector quantization (RVQ) codebooks (one per scale), and a phase- and amplitude-aware loss for training. The tokenizer aims to enable high-fidelity reconstruction across frequency bands (delta to gamma) for generative masked modeling. Evaluated on reconstruction error and four downstream BCI tasks (e.g., emotion recognition, seizure detection), it claims up to 15% accuracy gains over existing LBMs like LaBraM, BIOT, and NeuroGPT, positioning it as a foundation for neural decoding and multimodal integration.

**Strengths:**

The paper addresses a timely challenge in EEG foundation models: improving tokenization to capture multi-scale frequency dynamics, which is crucial for noisy, complex brain signals. The multi-scale convolution and RVQ design incorporate domain knowledge (e.g., EEG frequency bands), potentially aiding reconstruction fidelity. The phase/amplitude loss is a thoughtful addition grounded in signal processing. Experiments cover reconstruction metrics and downstream tasks, with claims of superior performance, and the work highlights scalability for LBMs.

**Weaknesses:**

Novelty is marginal: The tokenizer builds directly on RVQ from VQ-VAE (Esser et al., 2020) and multi-scale convolutions common in EEG models (e.g., EEGNet's depthwise convolutions for bands; Lawhern et al., 2018), without introducing new architectural or theoretical elements—e.g., the hierarchical codebooks per scale are a straightforward extension, and the loss is similar to phase-aware objectives in audio (e.g., EnCodec).

**Questions:**

1. The RVQ uses one codebook per frequency scale—how was the number of scales (e.g., delta-gamma) chosen, and what happens if bands overlap or vary by task?

2. Reconstruction error is lower than baselines, but does this translate to better perceptual quality?

3. Downstream gains are up to 15%—are these consistent across subjects/datasets, or driven by specific ones? Details on variance and p-values would help assess robustness.

4. The model is for generative masked modeling, but how does it compare to non-codebook tokenizers (e.g., BIOT's transformer) in masking ratios or pretraining efficiency?

5. For broader impact, test on non-EEG biosignals (e.g., ECG)—does the multi-scale approach generalize, or is it EEG-specific?

I will consider raising my score if all questions and concerns are solved.

---

> ### Author Response · Authors · 2025-11-19
> **Response to Reviewer gbAW**
>
> We sincerely thank the reviewer for their thoughtful review and valuable feedback.
>
> **Weaknesses** section:
>
> Regarding the reviewer's comment that our work does not introduce any new architectural or theoretical elements, we respectfully disagree. Our work solves an open problem in BCI foundation models: high-fidelity EEG tokenization. Indeed, our solution puts together well-established components (e.g., RVQ), but their integration is non-trivial; and it also introduces new ones (e.g., our loss function). Together, they make a state-of-the-art EEG tokenizer that enables accurate signal reconstruction.
>
> First, EEGNet uses depthwise convolutions to achieve spatial mixing across EEG channels, not frequency bands. On the contrary, our method incorporates multi-scale convolutional kernels (similar to an Inception-style architecture) to effectively capture a broader range of frequency bands. This is essential because existing codebook-based tokenizers (e.g. LaBraM) rely on a single kernel size in the encoder, which limits their ability to represent multi-scale EEG dynamics.
>
> Second, this is the first time RVQs are used to capture fine-grained EEG signal elements per scale, improving reconstruction fidelity beyond what single-codebook quantizers (e.g LaBraM) can achieve.
>
> Finally, our training objective, derived from well-established signal-processing principles, enables accurate signal reconstruction. The use of amplitude log-loss, sinusoidal (sine/cosine) components and a unit-circle penalty are, to our knowledge, introduced for the first time in the context of EEG signal reconstruction. These loss components are fundamentally different from those in frameworks such as EnCodec, which rely on discriminator and spectrogram losses (not present in our work) rather than signal motivated constraints.
>
> **Responses to Reviewer's Questions**
>
> **Q1**
> The selected frequency bands are grounded in well-established neuroscientific findings: brainwave activity is known to decompose into distinct frequency bands, each associated with different cognitive and physiological processes and task-dependent activation patterns (e.g., see [1]). All EEG recordings contain each of the frequency bands, though different tasks elicit increased activity in certain bands or combinations of bands. Our method is designed to simultaneously disentangle and represent these relevant bands, enabling the model to learn their unique characteristics in a principled manner.
>
> [1] Newson JJ, Thiagarajan TC. EEG Frequency Bands in Psychiatric Disorders: A Review of Resting State Studies. Front Hum Neurosci. 2019;12:521.
>
> **Q2**
> Our primary objective is to develop a high-fidelity tokenizer capable of accurate signal reconstruction. To achieve this, we must predict all frequency bands rather than limiting the model to those associated with specific tasks. Therefore, we do not differentiate between tasks that might rely on fewer bands; our focus remains on comprehensive reconstruction accuracy. The proposed tokenizer captures frequency-specific dynamics, thus providing richer representations than existing approaches and, therefore, lower reconstruction error than SOTA (see Tables 1-3). Tables 4 and 9 show consistently enhanced downstream performance across diverse datasets and tasks compared to other larger foundation models.
>
> **Q3.**
> For the evaluation datasets we used the datasets based on the [*] benchmark: High Gamma (14 subjects),  Pavlov 2022 (65 subjects), OPENBMI-ERP (54 subjects), SLEEP-EDF (78 subjects) and Physionet-Eyes (103 subjects). We performed the same subject-independent cross-validation across the fine-tuning of all models. Table 4 also reports the variance between folds to highlight robustness. During the review period, we have also added results for an additional task, ERP detection, which further supports the generalization and practical value of our approach.
>
> | Model     | ERP               |
> |-----------|-------------------|
> | NeuroGPT  | 0.757 ± 0.048     |
> | CBraMod   | 0.777 ± 0.052     |
> | BIOT      | 0.500 ± 0.000     |
> | LaBraM    | 0.822 ± 0.040     |
> | EEGPT     | 0.668 ± 0.146     |
> | **NeuroRVQ** | **0.876 ± 0.033** |
>
> [*] Na Lee et al. Assessing the Capabilities of Large Brainwave Foundation Models. In 2025 IEEE 35th International Workshop on Machine Learning for Signal Processing (MLSP) and ICLR2025 Workshop on Spurious Correlation and Shortcut Learning: Foundations and Solutions, 2025.

---

> > ### Author Response · Authors · 2025-11-19
> > **Response to Reviewer gbAW (2)**
> >
> > **Q4.**
> > In terms of masking, we follow similar approaches to other baseline models. Specifically, we use random patch masking (across EEG channels and time) with masking ratio of 0.5 similar to codebook-based models (e.g, LaBraM) and non-codebook (e.g., CBraMod and BIOT). As seen in Table 9, our model has both small backbone and classification heads. For fair comparison, we fine-tuned all pretrained models and evaluated their downstream task performance using identical experimental conditions.
> >
> >
> > **Q5.**
> > We focus on EEG signal reconstruction only. Preliminary experiments indicate that our approach is transferable to other biosignals (e.g., EMG). Reporting such results is beyond the scope of this paper.
> >
> > **Conclusion**
> > We hope the reviewer recognizes the potential impact of this work as a meaningful and timely contribution to the ICLR community and the great benefits of EEG tokenization important to the BCI Foundation Model research community.

---

> > > ### Author Response · Authors · 2025-11-26
> > > **Follow-up to Reviewer gbAW**
> > >
> > > We would like to kindly follow up regarding our rebuttal response, as we have not yet received further comments from the reviewer. If there are any additional questions or points requiring clarification, we would be happy to address them.

---

> > > > ### Author Response · Authors · 2025-12-01
> > > > **Final Comment to Reviewer gbAW**
> > > >
> > > > We thank the reviewer for their feedback. We hope that our responses satisfactorily address the reviewers’ questions and concerns.
> > > >
> > > > **Downstream Task Performance**
> > > > In our updated manuscript, we have also added a newly released Motor EEG Foundation Model, MIRepNet. The final performance in downstream tasks can be summarised as follows:
> > > >
> > > > | Model       | Motor        | ERP          | Memory        | Sleep      | Eyes         | Mean         |
> > > > |------------|-------------|-------------|--------------|-------------|-------------|-------------|
> > > > | NeuroGPT    | 0.682±0.083  | 0.757±0.048 | **0.597±0.029** | 0.674±0.033 | 0.827±0.036 | 0.707±0.046 |
> > > > | CBraMod     | 0.614±0.104  | 0.777±0.052 | 0.574±0.038 | 0.635±0.041 | 0.839±0.041 | 0.688±0.055 |
> > > > | BIOT        | 0.443±0.079  | 0.500±0.000 | 0.510±0.018 | --          | 0.763±0.049 | --          |
> > > > | MIRepNet    | 0.689±0.086  | --          | --          | --          | --          | --          |
> > > > | LaBraM      | 0.630±0.076  | 0.822±0.040 | 0.526±0.026 | 0.652±0.037 | 0.799±0.047 | 0.686±0.045 |
> > > > | EEGPT       | 0.313±0.035  | 0.668±0.146 | 0.520±0.017 | 0.634±0.044 | 0.797±0.037 | 0.587±0.056 |
> > > > | **NeuroRVQ** | **0.700±0.073** | **0.876±0.033** | 0.574±0.027 | **0.728±0.028** | **0.869±0.026** | **0.749±0.037** |
> > > >
> > > > **Other Biosignal Modalities Tokenization**
> > > > Although this was not part of the original scope of the paper, in response to the reviewer’s suggestion regarding potential extension to other biosignal modalities, we conducted additional ablation studies on EMG and ECG data. Our experiments show that NeuroRVQ can be naturally extended to these modalities, achieving high-fidelity tokenization and accurate signal reconstruction. The results, presented in Figures 7–8 in Appendix H, clearly demonstrate the effectiveness of NeuroRVQ beyond EEG and highlight that the underlying principles of our tokenizer generalize well to other biosignals. These findings open the door to broader applications of foundation models across the biosignal domain, **making NeuroRVQ the first generic model to enable high-fidelity biosignal tokenization and, consequently, precise reconstruction.**

---

### Author Response · Authors · 2025-12-01
**General Comment for AC's decision-making (1/3)**

We sincerely thank all reviewers for their thoughtful evaluations and constructive feedback. Because the rebuttal discussion concluded abruptly and several scores reverted afterward, we believe it is crucial to clearly summarize for the Area Chairs the significance and contributions of our work as well as the substantive improvements made during the rebuttal phase.

**About the Reviewers** We actively engaged throughout the rebuttal discussion period, which led to substantial improvements to our work:

1. We provided detailed and accurate responses to all reviewer questions, which resulted in increased reviewer scores.
2. We incorporated reviewer suggestions, enabling us to demonstrate high-fidelity tokenization beyond EEG and extend our claims to other biosignal modalities.
3. We expanded our benchmarking experiments, further showcasing the superior performance of NeuroRVQ.
4. In response to a reviewer, we have prepared a repo to provide our models open-source (upon acceptance), with a demo functionality for users to see the tokenization capabilities.

**Although the rebuttal discussion ended abruptly, one reviewer had already significantly increased their score and another indicated they would substantially raise their score pending our responses. Unfortunately, due to the sudden conclusion of the discussion, we were unable to receive follow-up responses to the detailed answers we provided. As a result, some reviewers may not have seen our latest benchmarking results, that we will provide the models open-source or the new extension to additional biosignal modalities highlighted in our rebuttal updates. We respectfully ask the Area Chairs to take these factors into consideration when assessing our submission.**

**Contribution** Our work addresses a critical open problem in foundation models for Brain-Computer Interfaces (BCIs): achieving high-fidelity EEG tokenization. The key to this goal lies in accurately modeling the full spectrum of neural oscillations. Our tokenizer is grounded in well-established neuroscientific principles demonstrating that EEG signals decompose into distinct frequency bands, each linked to specific cognitive and physiological processes with task-dependent activation patterns. All EEG signals inherently contain these bands, and our method is designed to disentangle and represent them simultaneously, enabling the model to learn their unique characteristics in a principled manner. Because our primary objective is accurate reconstruction, we predict all bands rather than restricting the model to task-specific subsets, yielding comprehensive and high-fidelity representations. **NeuroRVQ is the first model to enable high-fidelity EEG tokenization and, consequently, accurate signal reconstruction.**

**Tokenization** In the paper, we demonstrate substantially improved reconstruction fidelity compared to the current SOTA tokenizer, namely LaBraM. We conduct a series of experiments comparing both the original open-source model and a version trained from scratch on the same datasets as NeuroRVQ, enabling a fair head-to-head evaluation, as shown in Tables 1-3. For example, below we report the out-of-distribution mean squared error (MSE) reconstruction performance on motor tasks across frequency bands, comparing NeuroRVQ against both LaBraM versions (original pre-trained and ours):

|                | Raw Signal | Delta   | Theta   | Alpha   | Beta    | Gamma   |
|----------------|-----------|--------|--------|--------|--------|--------|
| LaBraM (orig.) | 2.004     | 0.863  | 0.326  | 0.400  | 0.350  | 0.064  |
| LaBraM (ours)  | 1.893     | 3.313  | 0.669  | 0.721  | 0.339  | 0.049  |
| **NeuroRVQ**   | **0.090** | **0.032** | **0.010** | **0.017** | **0.038** | **0.009** |

While MSE is one metric, Figure 4 captures qualitatively the superior performance and the best high-fidelity reconstruction capabilities of NeuroRVQ.

**Other Biosignal Modalities Tokenization** Although this was not part of the original scope of the paper, in response to a reviewer’s suggestion regarding potential extension to other biosignal modalities, we conducted additional ablation studies on EMG and ECG data. Our experiments show that NeuroRVQ can be naturally extended to these modalities, achieving high-fidelity tokenization and accurate signal reconstruction. The results, presented in Figures 7–8 in Appendix H, clearly demonstrate the effectiveness of NeuroRVQ beyond EEG and highlight that the underlying principles of our tokenizer generalize well to other biosignals. These findings open the door to broader applications of foundation models across the biosignal domain, **making NeuroRVQ the first generic model to enable high-fidelity biosignal tokenization and, consequently, precise reconstruction.**

---

> ### Author Response · Authors · 2025-12-01
> **General Comment for AC's decision-making (2/3)**
>
> **Downstream Task Performance** While the codebook-based state-of-the-art tokenizer is the central motivation of the paper, we also showcase that high-fidelity tokenization directly links to state-of-the-art downstream task performance. To this end, we train the NeuroRVQ foundation model that operates on the tokenized representation learned by the NeuroRVQ tokenizer, using a masked-token prediction strategy in which a subset of input patches is randomly masked. This objective encourages
> the network to infer missing tokens from their surrounding context. We evaluate NeuroRVQ against a range of open-source EEG foundation models in a recently published EEG Foundation Model benchmark [*], which is motivated by causal reasoning and the removal of task-discriminative artifacts. To ensure fairness, our pre-training dataset does not include any of the benchmark data, whereas some competing baselines do. During the review rebuttal period, we have also added results for the last task of this benchmark (ERP detection task), which further supports the generalization and practical value of our approach. We also added a newly released model, MIRepNet [1], in the motor task since it is a motor EEG foundation model. We performed the same subject-independent cross-validation across the fine-tuning of all models and display the variance between folds to highlight robustness.
>
> | Model       | Motor        | ERP          | Memory        | Sleep*       | Eyes         | Mean         |
> |------------|-------------|-------------|--------------|-------------|-------------|-------------|
> | NeuroGPT    | 0.682±0.083  | 0.757±0.048 | **0.597±0.029** | 0.674±0.033 | 0.827±0.036 | 0.707±0.046 |
> | CBraMod     | 0.614±0.104  | 0.777±0.052 | 0.574±0.038 | 0.635±0.041 | 0.839±0.041 | 0.688±0.055 |
> | BIOT        | 0.443±0.079  | 0.500±0.000 | 0.510±0.018 | --          | 0.763±0.049 | --          |
> | MIRepNet    | 0.689±0.086  | --          | --          | --          | --          | --          |
> | LaBraM      | 0.630±0.076  | 0.822±0.040 | 0.526±0.026 | 0.652±0.037 | 0.799±0.047 | 0.686±0.045 |
> | EEGPT       | 0.313±0.035  | 0.668±0.146 | 0.520±0.017 | 0.634±0.044 | 0.797±0.037 | 0.587±0.056 |
> | **NeuroRVQ** | **0.700±0.073** | **0.876±0.033** | 0.574±0.027 | **0.728±0.028** | **0.869±0.026** | **0.749±0.037** |
>
> (BIOT could not be tested on Sleep since the benchmark electrodes are missing from the pre-trained model.)
>
> | Model      | Backbone | Motor Head   | Memory Head  | Sleep Head  | Eyes Head   |
> |-----------|----------|-------------|-------------|------------|------------|
> | CBraMod   | 4.9M     | 50,081,804  | 40,481,201  | 73,207,406 | 41,121,201 |
> | NeuroGPT  | 79.5M    | 270,756     | 270,657     | 270,822    | 270,657    |
> | BIOT      | 3.2M     | 1,028       | 257         | -          | 257        |
> | LaBraM    | 5.8M     | 804         | 201         | 1,206      | 201        |
> | EEGPT     | 25.7M    | 260         | 65          | 390        | 65         |
> | **NeuroRVQ** | 5.9M  | 3,204       | 801         | 4,806      | 801        |
>
> **As shown above, NeuroRVQ achieves the best or next-best performance in each task and the highest overall mean accuracy across the full benchmark - despite having only a fraction of parameters compared significantly larger foundation models. This combination of strong generalization and compact size highlights the effectiveness of our high-fidelity NeuroRVQ tokenizer and the representations it enables.**
>
>
> [*] Na Lee et al. Assessing the Capabilities of Large Brainwave Foundation Models. In 2025 IEEE 35th International Workshop on Machine Learning for Signal Processing (MLSP) and ICLR2025 Workshop on Spurious Correlation and Shortcut Learning: Foundations and Solutions, 2025.
>
> [1]: Dingkun Liu et al. Mirepnet: A pipeline and foundation model for eeg-based motor imagery classification, 2025

---

> > ### Author Response · Authors · 2025-12-01
> > **General Comment for AC's decision-making (3/3)**
> >
> > **Novelty** While our approach leverages several well-established components (e.g RVQ utilization), their integration is non-trivial, and we introduce novel elements—such as a new loss function—that are essential to its success.
> >
> > Regarding the training loss: Our training objective, derived from well-established signal-processing principles, enables accurate signal reconstruction. The use of amplitude log-loss, sinusoidal (sine/cosine) components and a unit-circle penalty are, to our knowledge, introduced for the first time in the context of EEG signal reconstruction. These loss components are fundamentally different from those in frameworks such as EnCodec, which rely on discriminator and spectrogram losses (not present in our work) rather than signal motivated constraints.
> >
> > Regarding the RVQ utilization: This is the first time RVQs are used to capture fine-grained EEG signal elements per scale, improving reconstruction fidelity beyond what single-codebook quantizers (e.g LaBraM) can achieve. While two other works, BrainOmni and BrainCodec, both use RVQ codebook, they exhibit significant limitations in EEG reconstruction fidelity. BrainOmni uses an RVQ codebook to separate different source components, while NeuroRVQ utilizes RVQ codebooks to capture signal components. BrainCodec is a direct application of EnCodec to the EEG / iEEG domains and relies on a single RVQ codebook and discriminator. Unlike these works, NeuroRVQ employs multiple RVQ codebooks per frequency band to capture fine-grained signal elements.
> >
> > The significant improvements in EEG reconstruction arise not simply from the use of RVQ itself, but from the combination with the carefully designed training loss as well. A qualitative and quantitative comparison further illustrates the difference between our work. As shown in Figure 4 of BrainOmni and Figures A9 and A16 of BrainCodec, these models do not recover high-frequency activity and fine-scale structural details. In contrast, our results in Figure 4 and Tables 1–3 demonstrate substantially improved reconstruction fidelity, addressing this known limitation. Together, these contributions result in a state-of-the-art EEG tokenizer that advances the current capabilities of BCI foundation models.
> >
> > **Conclusion**
> >
> > NeuroRVQ sets a new benchmark for high-fidelity EEG tokenization, achieving unprecedented reconstruction accuracy while enabling state-of-the-art performance across diverse downstream tasks. Its novel multi-codebook RVQ design and carefully crafted training loss not only advance EEG modeling but also generalize to other biosignal modalities, opening new directions for foundation models in neuroscience and biomedical AI. With strong generalization, compact size and demonstrable superior performance, NeuroRVQ (released open-source upon acceptance) represents a timely and impactful contribution to the ICLR community. We hope the AC recognizes the potential impact to the field.

---

### Meta-Review · Area_Chair_xZoD · 2026-01-07

**Summary:**

### Strengths

- Domain-Informed Design: NeuroRVQ leverages neuroscientific knowledge of EEG frequency bands to design a multi-scale convolutional tokenizer, aligning with known brainwave dynamics.

- Strong Performance: NeuroRVQ achieves lower reconstruction error and higher classification accuracy (up to 15% improvement) compared with existing EEG foundation models such as LaBraM, BIOT, and NeuroGPT.

- Comprehensive Evaluation: Extensive experiments on multiple benchmark datasets demonstrates the model’s robustness and cross-task consistency.

### Weaknesses

- Marginal Novelty and Incremental Contribution: Despite solid engineering, the architectural innovation is limited. The use of multi-scale convolutions, residual vector quantization (RVQ), and phase/amplitude-aware losses are all adaptations of existing techniques from EEGNet, VQ-VAE, and EnCodec. The integration of these known methods, while practical, does not constitute a substantive theoretical or architectural advance.

**Reviewer Concerns:**

Most of the empirical and implementation-related concerns raised by the reviewers have been effectively addressed. The authors clarified the selection of EEG frequency bands with neuroscientific justification, added detailed dataset statistics including subject counts, and provided additional downstream results such as ERP detection and comparisons with MIRepNet. They also offered clear explanations for hyperparameter choices, demonstrated stable training without overfitting despite high epoch counts, and included preliminary experiments showing the model’s applicability to EMG and ECG signals.

However, the key conceptual and theoretical concerns remain unaddressed. Reviewers gbAW and n4Y3 maintain that the paper lacks genuine novelty, as its main components are adaptations of well-established techniques rather than fundamentally new ideas. Despite improved empirical coverage, the work still falls short of demonstrating a substantive theoretical contribution or clear innovation beyond prior EEG foundation models.

**Reviewer Scores:**

Reviewer maKs committed to increasing score. Although most concerns had been addressed, and that other reviewers might also consider raising theirs. However, the paper still remains marginally below the acceptance threshold overall.

---

### Decision · Program_Chairs · 2026-01-26

Reject